

# Assessing future impacts of tropical cyclones on global banana production

Sophie Kaashoek[1], Žiga Malek[1], Nadia Bloemendaal[1,2], Marleen C. de Ruiter[1],

[1]Institute for Environmental Studies, Vrije Universiteit Amsterdam, De Boelelaan 1111, 1081HV, Amsterdam, the Netherlands
[2] Royal Netherlands Meteorological Institute (KNMI), Utrechtseweg 297, 3731 GA De Bilt, The Netherlands

*Correspondence to*: Marleen de Ruiter (m.c.de.ruiter@vu.nl)

**Abstract.** Tropical cyclones (TCs) are projected to increase in intensity globally, impacting human lives, infrastructure, and important agricultural activities, such as banana production. Banana production is already impacted by TCs in several parts of the world, leading to price volatility and impacted livelihoods of banana producers. While many potential impacts on banana

production have already been quantified on a local scale, it remains unclear how bananas could be impacted by TCs across the globe under future climate conditions. To address this, we first looked at the documented impacts of cyclones on banana production from different places all around the world. Using spatially explicit data on banana producing regions and future TC occurrence and magnitude, we then identify the spatial distribution and extent of areas where TCs could impact banana production. Our results suggest that considerable portions of global banana production are at risk to be impacted by TCs under

future climate conditions, and we show this for different return periods (RP).

Globally, at the 100-year return period, 24.3 % of all bananas producing areas will be majorly or completely damaged under present climate conditions and 26.5% under future climate conditions. When looking at production, we see that 30.1% of global production is projected to be majorly or completely damaged at the 100-year RP. The regions predominantly affected by future TCs are Asia and the Caribbean, experiencing substantial disruption in banana production. Our results therefore

indicate, that considerable efforts in climate change mitigation and adaptation need to be made in order to ensure the stability of global banana supply chains.

## 1. Introduction

Currently, the global surface temperature has already increased between 0.8 and 1.3 degrees Celsius above pre-industrial levels (Coumou, 2012). These increasing global temperatures trigger fast changing weather patterns showing more frequent and

intense extremes affecting water and food security with high confidence (IPCC, 2021). Particularly tropical cyclones (TCs) are projected to increase in intensity (Knutson et al 2020). This is predominantly driven by high sea-surface temperatures (SSTs), which serve as fuel for an intensifying TC. This is an alarming projection, since TCs can substantially impact coastal communities, causing potentially large agricultural and financial losses and human casualties. These categories represent cyclones with the most extreme intensities (Mendelsohn, 2012). However, the frequency of a TC- event occurring is expected



to decrease. This is due to a decrease in the strength of the Walker Circulation and a trend towards a period with more El Nino events (Beer, 2014). Due to a higher sea surface temperature, there is faster evaporation with more energy extraction out of the water. Consequently, more energy is released by condensation of the particles resulting in an increase in category 4 or 5 events. The development of the TC will happen more efficiently making it easy for the TC to increase in speed. Also, the water holding capacity increases due to higher CC, generating more intense rainfall (Coumou, 2012).

One of the agricultural activities with a high socio-economic significance that is most under threat of TCs, is banana production. A reduced availability of bananas leads to higher prices for consumers and more worrisome, to a decrease in income and economic prosperity in exporting countries (FAO, 2022). Due to our globalized economic system and its strong interdependencies, the potential negative impact on the production system will not only be harmful on a local scale, but will rather become a global issue (Lesk, 2016). Consequently, the partial or full destruction of banana plantations across larger

areas would have a large influence on the currently stable and reliable global value chain of bananas (Varma, 2019). Banana production is already expected to be more impacted by numerous extreme weather events in the coming decades such as drought, heavy precipitation, and heatwaves (Malek et al. 2022). Yet so far, studies on the impact of extreme weather on agricultural areas have mainly focused on the influence of extreme heat or precipitation deficit on production, thereby lacking a perspective on high- impact low-probability events like TCs (Calberto, 2015, Lesk, 2016, Varma, 2019, Malek et al, 2021).

This is remarkable, since the direct impact of TCs is largest on the agricultural sector (FAO, 2015, Kunze, 2021). Research has shown that there is a clear negative direct impact of TCs on banana producing areas (Huigen, 2006, Robinson, 2010, Mohan, 2017). TCs can completely disrupt the banana production area and it can take up to a year to recover depending on farmers' access to facilities and finance (Mamuye, 2016). For example, the 2021 Hurricanes Eta and Iota caused a 51% decrease of banana yields in Honduras (FAO, 2022. P.2). Due to production disruption, there is less supply increasing the average price

of the fruit (Beer, 2013). However, due to a lack of available data on TC projections, the spatial impact of this hazard for bananas could not be identified on a global scale (Varma, 2016, Malek, 2021). Several indirect risk assessments have been performed occurring in the time outside the TC event on macro- or meso-scale (Huigen, 2006, Beer, 2013, Mohan, 2017). It is expected that in the coming decades banana producers in Southeast Asia, Oceania and the Caribbean will likely be impacted by TCs, but there is no evidence of where impacts are on a supra-national scale (Cavendish, 2015, Varma, 2019, Malek, 2021).

This is problematic, as banana cultivators have no reliable projections of the probability to be affected by a TC. To many rural households, bananas generate an important part of their income (Mohan, 2017). In addition, bananas, together with plantain, are an important source of nutrition for millions of people. When banana producers are provided with better estimates of the likelihood and location of these indices, they will be able to increase their resilience against these devastating disruptions by mitigation or adaptation (Chavez, 2015). A disruption in banana production will negatively impact the United Nation's

Sustainable Development Goals, including the first (no poverty), second (zero hunger) and third (good health and well-being) goals (UN, 2015).



In this paper, we therefore assess how future TCs could impact banana production on a global scale. We first synthesize the documented impacts of TCs on banana production, by reviewing recent studies. Using data on the global distribution of current and future TCs, we identify where banana producing areas will be exposed to TCs. Finally, we quantify and map the extent of banana production that will experience more frequent and damaging TCs. This way, we contribute to the ongoing efforts to better understand and mitigate impacts of TCs on agriculture.

## 2. Methods

### 2.1 Methods overview

We perform a risk assessment by mapping the global TC risk to banana producing areas under present day (1980-2017) and future (2015–2050 under SSP585 forcing scenario) climate conditions (Figure 1). We follow the risk framework of the UNDRR (2017) that is based on the understanding that disaster risk is the product of hazard, exposure, and vulnerability. We refer to hazard as the probability of occurrence of a certain TC wind speed, exposure as the banana producing area at risk, and vulnerability as the fraction of banana plants that will be damaged by a natural hazard (UNDRR, 2017).

First, we performed a vulnerability assessment through a systemic literature review on past TC events damaging banana plantations to identify the physical vulnerability of bananas to TCs. We identified 22 studies with quantified impacts of TCs on banana plantations around the world. We used these studies to calculate the damage function (vulnerability) estimating against what wind speed (in m/s) banana plantations are damaged. A damage function is a common tool used to present the vulnerability of assets, in this case banana producing areas, to hazard, in this case a TC (De Ruiter et al., 2017). Due to the lack of data on the relative susceptibility of all banana-producing areas globally, the relative damage ratio needs to be developed. The relative damage function is transferable in time and space and therefore independent of market values from the different geolocation (Merz, 2010). Consequently, we develop a vulnerability curve presenting the vulnerability of the exposed assets (banana producing areas) to the hazard characteristics. Subsequently, we map the hazard using the most recent publicly available data on TC characteristics for the current and future climate (Bloemendaal et al. 2020a, Bloemendaal et al. 2022). Consequently, we overlay the processed TC data with data on the global distribution of banana production (Yu et al. 2020, IFPRI 2019), to map the damage distribution. This enabled us to identify the global spatial distribution of banana producing areas at risk to the future damage from TC under current and SSP585 climate conditions.

### 2.2 Tropical cyclone hazard mapping

To represent the hazard component of our risk framework, we use the Synthetic Tropical cyclOne geneRation Model (STORM) present and near-future climate TC wind speed return period (RP) maps (Bloemendaal et al 2020b, Bloemendaal et al 2022).





STORM is the first model that captures future global-scale TC wind speed probability trends derived at high (10 km) spatial resolution. STORM is created using a synthetic, fully statistical modelling approach that simulates new TCs by using statistical information on TC characteristics (frequency, track, and intensity) derived from historical observations (1980-2017)  STORM

then simulates 10,000 years of synthetic TCs based on present-day climate conditions (1980-2017), thereby providing the necessary data for the calculation of wind speed RPs (in m/s). To analyze changes in TC risk under climate change, we use the STORM future climate dataset (Bloemendaal et al 2022). This dataset extracts future-climate TC statistics using four high resolution GCM climate models based on present and Shared Socioeconomic Pathway 585(SSP) 2015-2050 climate conditions and the STORM algorithm. SSP585 is a high-emissions scenario and assumes that there will be no mitigation policies on

greenhouse gas emissions, leading to high radiative forcing (O'Neill et al. 2016). STORM projects solely the SSP585 high forcing scenario because there is little deviation in terms of TC activity between lower-forcing scenarios and the high emission SSP585 scenario (IPCC, 2021 report 43, Bloemendaal, et al. 2022). The STORM datasets are generated on a global scale, with the datasets split up over six basins (Eastern Pacific, North Atlantic, North Indian, South Indian, South Pacific, Western Pacific). The STORM present and future wind speed RP datasets provide 10-meter 10-minute maximum wind speeds at various

RP levels. From these datasets, we extract the 10-, 100-, and 200-yr wind speed RP maps, to account for both events with higher probabilities but lower intensity, and events with lower probabilities but higher intensities. Note that a high RP corresponds to a low probability of occurrence and thereby a higher risk probability (Ward, 2011, Bevacqua, 2019).

## 2.3 Exposure

### 2.2.1 Banana production and impacts of tropical cyclones

Globally, bananas are the most popular fruit (FAO, 2022). In countries all over the world with significantly different climates, bananas provide a large part of nutritional diversity in the diet of the average consumer. The wide-spread benefits of bananas include its nutritious values; they are rich in calories, are perfectly suitable for dietary diversity and they contain a variety of vitamins & fibers (Petsakos, 2019). A sustainable production of the fruit poses an important source for nutrition to maintain food security and sustain income (Petsakos, 2019, Varma, 2019). In Latin American and Caribbean (LAC) and Asian countries,

bananas remain a significant contributor to the economy and provide a livelihood for millions (FAO, 2019; Varma, 2019).

In 2021, global banana export volume was 20 million tonnes, of which Latin America & Caribbean (LAC) exported 16 million tonnes of bananas. The 5 largest exporting countries are Ecuador, the Philippines, Costa Rica, Guatemala, Colombia, and the Dominican Republic (FAO, 2022). The most consumed and traded bananas can be divided in two large groups; the 'dessert'

banana types which are sweet and can be consumed raw or 'cooking' banana types who need to be treated before use (FAO, 2019). Cooking bananas are mainly found in Asia and East Africa. The dessert banana is concentrated in Latin America and Asia, including the Cavendish type also in the Caribbean. The harvest of the plantain is found primarily in Africa, Latin



America and Asia. The estimated size of the global total banana plantation is 5.6 million ha (Machovina, 2013, Coltro, 2019, FAO, 2019). In this study, we assume all banana types exhibit the same vulnerability to TC wind speeds.


TCs can cause significant damage to banana producing areas in different ways (Table 1). Despite the diverse impacts on banana production, TCs particularly lead to loss through wind speeds, and less so by amount of rainfall (Lin 2020). This is mostly because the fruit easily blows off the plant by strong winds and directly creates crop losses. Humid winds can have a positive influence by helping to pollinate banana flowers, disperse pests and beneficial insects (Robinson, 2010). On the negative side,
strong winds can cause physical damage to the fruit itself, either destroying them, or reducing the quality of production, potentially limiting their direct human consumption.

**Table 1: Types of damage from wind to banana**

| Type of damage | Description | Source |
|---|---|---|
| Recovery period | The harvest period of bananas takes approximately a year, so when banana plantations are damaged the recovery period can take up to 9-12 months | Mamuye, 2016 |
| Height | Due to height of the plant more susceptible to TC compared to other crops | Huigen, 2006, Robinson, 2010 |
| Leaves changes production capacity | The producing capacity of the banana is reduced when leaves are damaged. | Mohan, 2017 |
| Productivity | The wind can also cause the fruit to fall off the plant prematurely or affect the productivity of the herb by influencing the layer between the undisturbed air adjacent to the surface and temperature of the leave | Robinson, 2010 |
| Low wind speed | Even wind at lower speeds (10 m/s) can cause damage to banana plants, particularly if the plants are already weakened or stressed for other reasons such as pests and diseases. In some cases, this weaker wind may be strong enough to uproot the plants or cause significant damage to the leaves and stems | Mohan, 2017 |
| Dry wind | Dry winds can lead to water shortage impacting growth. | Robinson, 2010 |





| Seasonal winds | Strong seasonal winds can cause leave tearing, and slower winds can increase leaf and dust abrasion. | Robinson, 2010 |
|---|---|---|
| High winds | In addition to the physical damage to the plants and fruit, high winds can also cause other issues that can impact banana production. For example, the winds can blow debris and sand around, which can make it more difficult for the plants to grow properly. The winds can also create dust storms, which can reduce visibility and make it harder for farmers to work in the fields | El-Kady, 2012 |

### 2.3.3 Mapping banana producing areas

To map the banana producing areas that will be susceptible to a TC, we used data on global spatial distribution for major crops, which includes bananas (IFPRI 2019, Yu et al. 2020). These data present the most recent systematically developed data on banana distribution, by combining a set of statistical and remote sensing data. The spatial resolution of the data is roughly 10x10 km at the equator. To remove areas with insignificant banana production, we only consider areas where bananas are grown on more than 5 ha within the 10x10km pixel. We then looked at banana production in tons in these areas, using the same data, as MAPSPAM not only provides data on the spatial distribution of harvested areas, but also production.

### 2.1 Vulnerability

### 2.3.1 Vulnerability curve

Vulnerability ascribes the characteristics of the exposed asset that make it susceptible to the effects of a certain hazard (here a TC) (UNDRR, 2017). The vulnerability curve assesses the damage state against the hazard intensity showing the wind speed (m/s) interval and the mean damage ratio (MDR) (Wanmei, 2016, Yum et al., 2021). Vulnerability curves can be based on empirical, analytical, and expert opinion or judgement methods (Merz, 2010). Yet so far, most academically developed vulnerability curves for TCs quantify wind speed damage focusing on the built environment (Sun, et al. 2021, Yum, et al. 2021). To the best of our knowledge, there is currently no evidence on vulnerability curves of wind-induced damage for bananas. To quantify the level of damage to banana plantations caused by a certain wind speed, we therefore have to develop vulnerability curves. This is done by executing a systemic review of reported past TC events damaging banana plantations. To construct the curves, we use an empirical approach similar to the research by Yum et al (2021) (see Figure 3). An empirical approach allows for a higher accuracy of the damage curve (Merz, 2010). Additionally, to reduce uncertainty, expert opinions are also used (Wanmei, 2016). To retrieve reports of past events of damaged banana plantation by wind speed, peer-reviewed articles, and news items about destroyed banana plantations by TC wind were explored by Google Search and the Google Scholar and WorldCat academic databases (published anytime using the following search terms: bananas + storms, hurricanes,





(tropical) cyclones, typhoons; and we excluded studies that did not include wind damages). The events were categorized into the time, location, windspeed and % of destroyed plantation. We identified 22 cases where impacts of TC on banana production were quantified from all around the globe (see S1). When an event had no mention of maximum wind speeds, we used the

historical TC database of JAXA/EORC (2023) and NASA (2022) and retrieved the wind speeds and trajectories for the affected region. This way we were able to identify the wind speed and trajectory of the TC at all locations. The 22 cases were categorized by time, location, windspeed and % of destroyed plantation.

### 2.3.2 Mean Damage Ratio

All 22 events found were used to calculate the damage ratio of the TC-induced damaged banana plantation against the corresponding wind speed intensity, similar to the method discussed by Yum et al. (2021). The damage ratio indicates the likelihood of a banana plant surpassing a certain damage level threshold (vmin) as the windspeed increases up to a certain critical point. Above the threshold, the exceedance probability increases before it reaches a 100% destroyed plantation (vmax), showing the plantation is fully destroyed (Yum et al., 2021). Yum et al. (2021) quantify the damage ratio based on insurance

data (using monetary values). In our study, we only assess the physical vulnerability due to the TC wind speed. The equation used is described in Eq. (1):

| | |
|---|---|
| $Damage\ ratio\ =\ \dfrac{X_{Destroyed}}{X_{Total}}$ | (1) |

Where $X_{destroyed}$ represents the destroyed plantation area (in ha), and $X_{total}$ represents the total plantation area (in ha). From

Equation (1), it follows that higher damage ratios relate to higher vulnerability. The formula is used for the events lacking a damage ratio. To quantify the damage ratio, the spatial destruction (ha destroyed) and the entire scope (ha plantation) needs to be identified. Some events referred to banana plantation losses based on total banana production area in a country. Given that countries have a significant difference in size and spatial distribution of production areas, it is necessary to assess the exact location where an individual TC struck the plantation. Hence, to obtain a robust and reliable database, events are removed

when the location could not be determined. 22 damage ratios (d1, d2, …., d22) were identified with corresponding wind speeds (v1, v2, … v22) (see S1 table S1.2 and table S1.3). The datasets were categorized from lowest to highest wind speed and grouped into three wind-speed intervals, namely: 0-25, 26-43, and above 44 (see Table 2). The wind speed intervals were established based on expert judgement (see Table 1) and the Saffir Simspon (1974) scale (see Table 2). The lower limit of the curve was found to be 10 m/s (denoted as vmin) and the upper limit was 44 m/s (denoted as vmax). According to Robinson

(2010) and Mohan (2017), evidence of damage to banana plantations was already observed at windspeeds of 10 m/s, which falls below the lower threshold of 26 m/s for classification as a TC (Simpson, 1974). Additionally, the events found in the database also show that moderate damage to a banana plantation below 26 m/s. To identify the banana producing areas at risk to wind speed, it was necessary to include the wind speed interval of 10-25 m/s in the analysis. Moreover, complete destruction





of a plantation was observed at 44 m/s in the database. Therefore, two wind speed intervals were determined as: 25-43 m/s
which indicated major damage, and 44 m/s and above which indicated complete damage.

Per wind speed interval, the associated damage ratios were divided by the total sum to calculate the "mean damage ratio"
(MDR). The mean damage ratio will allow for global heterogeneity around the 22 events and will enable this research to
apply a homogenous approach necessary for analysis on a supra-national scale (de Moel, 2013). The MDR (D) is calculated
using windspeed (v), and damage ratio (d) and are defined in Eq.(20). For example, for DV1 all the damage ratios with a
windspeed between 10-25 m/s from the past events were summed. Consequently, this sum was divided the total sum of all
damage ratios. The wind speed intervals with corresponding damage ratios are shown in Table 2.

| | |
|---|---|
| $$Dv1 = \left(\frac{(d1 + d2 + \cdots, d6)}{dx}\right) * 100$$ | (2a) |
| $$Dv2 = \left(\frac{(d7 + d8 + \cdots, d12)}{dx}\right) * 100$$ | (2b) |
| $$Dv3 = \left(\frac{(d13 + d14 + \cdots, d22)}{dx}\right) * 100$$ | (2c) |

**Table 2: Mean Damage Ratio per wind speed interval with corresponding damage level**

| Wind speed interval (10- min maximum average sustained windspeed) | Average damage | Saffir Simpson scale | Damage level |
|---|---|---|---|
| 0-10 | 0% | No | No Damage |
| 10-25 | **DV1**: 40% | No | Moderate damage |
| 26-43 | **DV2**: 79% | 1-3 | Major damage |
| 44> | **DV3**: 97% | 3-5 | Complete destruction |




### 2.3.3 Plotting the vulnerability curve

The vulnerability curve was created by plotting the windspeed intervals against the different damage ratios (Figure 1). The curve was developed using a cumulative distribution function. This is used to analyse the accumulated damage ratios. The shape of the curve follows that of similar vulnerability curves of damages caused by TC wind (e.g.Khanduri, 2003, Robinson, 2010, Mamuye 2016, Kim, et al. 2017, Yum, 2021).

**Figure 1: Constructed vulnerability curve of TC wind damages to bananas based on a meta-analysis of 22 studies, similar to Yum et al. (2021).**

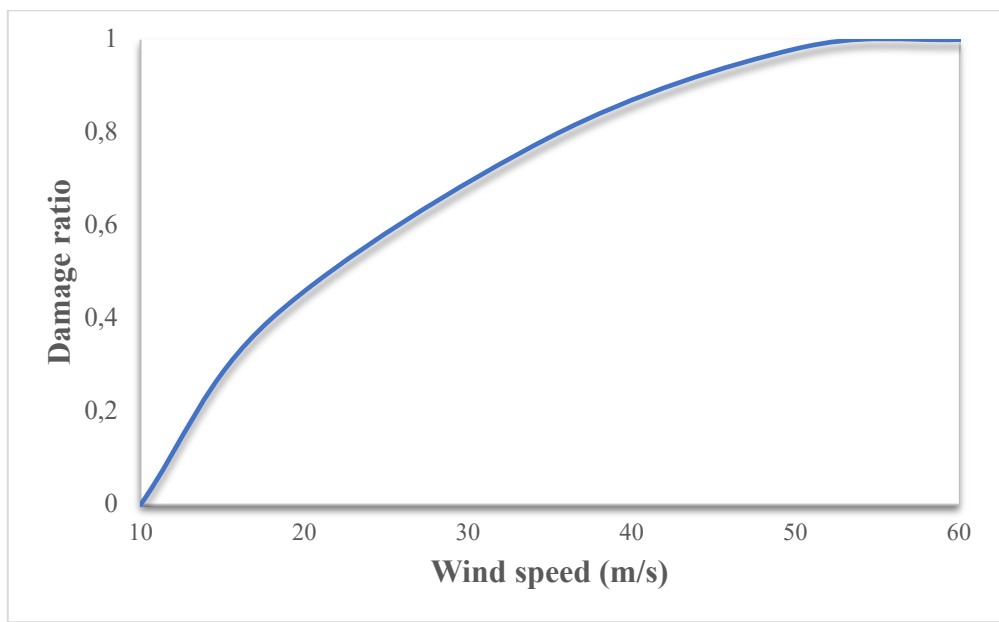

The global spatial distribution of TC induced damage to banana producing areas was mapped and calculated in GIS by overlaying the spatial distribution of banana production areas, the banana production (Yu et al. 2020, IFPRI 2019), and the STORM datasets (Bloemendaal, et al. 2020, Bloemendaal, et al. 2022). The STORM wind speed RP global raster datasets were calibrated based on the damage ratios found in the vulnerability assessments. The wind speed intervals of each RP-map were classified with the corresponding damage levels based on the results found in the vulnerability curve. The classification of the wind speed intervals was classified in accordance with labels of: 0-10 m/s (grey) being no damage (MDR = 0%); 10-25





m/s (yellow) being moderate damage being moderate damage (MDR = 40%); the third class 26-43 m/s being major damage (MDR = 79%); and the last class 44> m/s (red) being total destruction (>97 %).

## 3. Results

### 3.1 Banana producing areas

We first provide the spatial distribution of potential TC impacts on banana producing areas, considering both the current climate (1980-2018) and the climate conditions projected under SSP585 (2015-2050) by future TC's across four distinct Return Periods (RPs) (Figure 2). In addition, we identify the portion of damaged banana producing areas by future TCs for the current and future climate conditions under different RPs (Table 3). We observe an increase in the portion of damaged areas and a corresponding increase in damage levels for the three distinct return periods: 10 (RP10), 100 (RP100) and 200-year RP period

(RP200). s. Globally, at the 10-year RP, 16.5 % of all bananas producing areas will be moderately or majorly damaged under present climate conditions. At the 100-year RP, 34.5 % will be moderately, majorly, or completely damaged and 39% at the 200-year RP. Furthermore, it is evident that there will be a rise in the extent of areas susceptible to damage from future TCs under the SSP585 scenario (Table 4). At the 10-year RP, 11% of banana areas globally will be subject to increase moderately, and 9.5% to major destruction in the future, presenting a 8% and 51% increase compared to the current climate respectively

(Tables 3 and 4, Figure 2). In the SSP585 at the 100-year RP 21.7% of areas will be subject to major damages, and 4.8% to complete destruction, presenting a 3% and 50% increase compared to present climate conditions (Tables 3 and 4, Figure 2).






**Figure 2: Global distribution of impacted banana producing areas across different damage levels under various return periods for both current (1980-2018) and future climates (2015-2050). High resolution images for individual return periods are provided in S2.**

**Table 3: The values presented in the table are in percentage and represent the global distribution of impacted banana producing areas across different damage levels under various return periods for both current (1980-2018) and future climates (2015-2050).**


| | Present | | | Future | | |
|---|---|---|---|---|---|---|
| | RP10 | RP100 | RP200 | RP10 | RP100 | RP200 |
| Moderate | 10.2 | 10.2 | 10.5 | 11.0 | 12.2 | 13.0 |


| | | | | | | |
|---|---|---|---|---|---|---|
| Major | 6.3 | 21.1 | 22.3 | 9.5 | 21.7 | 23.1 |
| Complete | 0.0 | 3.2 | 6.3 | 0.0 | 4.8 | 7.5 |

**Table 4: The values presented in the table are in percentage and represent the increase of globally damaged banana producing area comparing current (1980-2018) and future climates (2015-2050) across different damage levels under various return periods.**

| | RP10 | RP100 | RP200 |
|---|---|---|---|
| Moderate | 7.7 | 19.8 | 23.4 |
| Major | 50.6 | 2.8 | 3.6 |
| Complete | | 50.1 | 18.6 |


We show that there are large disparities across the globe when it comes to banana producing areas being affected by TCs (Table 5). At RP100, in the Caribbean, 56.9% is completely damaged, while in South Asia, this figure stands at 0.5% under present climate conditions. The regions most prominently impacted by future TCs are Asia and the Caribbean, witnessing considerable harm inflicted upon their banana areas. South America, West and South Africa, will experience minimal impact

as TC's are rarely observed in these regions (Table 5).

**Table 5: The values presented in the table are in percentage and represent the impacted banana producing area in different regions across major and complete destructed levels under the RP100 for both current (1980-2018) and future climates (2015-2050). This table focuses on selected banana regions. However, not all regions are affected by tropical cyclones (TCs). The presented numbers in**
**the table represent shares or proportions rather than absolute values. To provide a more comprehensive understanding, absolute numbers in hectares can be found in S2 as well as the % damaged banana producing area per countries.**

| | Major | | Complete | |
|---|---|---|---|---|
| Region | Present | Future | Present | Future |
| Caribbean | 40.5 | 44.3 | 56.9 | 53.2 |
| Central America | 38.0 | 24.4 | 0.0 | 2.9 |
| South America | 1.8 | 1.9 | 0.0 | 0.0 |
| West Africa | 0.3 | 0.3 | 0.0 | 0.0 |
| East Africa | 29.5 | 30.4 | 0.0 | 3.4 |
| South Africa | 0.0 | 0.9 | 0.0 | 0.0 |
| Middle East and North Africa | 12.3 | 16.7 | 0.0 | 0.0 |
| East Asia | 66.8 | 61.7 | 2.7 | 10.7 |
| South Asia | 39.6 | 38.1 | 0.5 | 0.0 |
| Southeast Asia | 24.5 | 29.9 | 8.0 | 10.9 |
| Oceania | 21.0 | 21.2 | 0.0 | 5.2 |



## 3.2 Banana production

Impacts on production show even greater disparities (Table 6), as currently the productivity of different banana production
systems varies greatly across the globe. Furthermore, it can be concluded that there is an increase in impacted banana
production under the SSP585 scenario. We can observe an increased share of damaged areas and an increase in damage level
for the 3 different return periods, namely RP10, RP100 and RP200. At the RP200 the global share of major and complete
destructed banana production increases, but the total amount of impacted banana production decreases.

**Table 6: The values presented in the table are in percentage and represent the global distribution of impacted banana production**
**across different damage levels under various return periods for both current (1980-2018) and future climates (2015-2050).**

| | Present | | | Future | | |
|---|---|---|---|---|---|---|
| | RP10 | RP100 | RP200 | RP10 | RP100 | RP200 |
| Moderate | 8.5 | 14.9 | 12.5 | 10.2 | 16.8 | 11.7 |
| Major | 7.8 | 27.1 | 32.9 | 10.1 | 27.2 | 34.9 |
| Complete | 0.0 | 3.0 | 5.2 | 0.0 | 4.0 | 5.1 |

We found out, that the regions predominantly affected by TCs are Asia and the Caribbean, experiencing substantial disruption
in banana production. Banana production will be subject to complete damage in many parts of the Caribbean, totally amounting
to 58.4% of production under the future climate (although this presents a decrease compared to the current climate, where it
amounts to (68.1%). East and Southeast Asia, hosting considerable shares of global banana production, are projected to
experience 13.3% and 8.8% complete damage respectively. Despite other areas experiencing minor shares of production being
impacted by complete damage, many areas are projected to experience major damages un more than 20% of banana production
under the future climate: East, South and Southeast Asia, Caribbean and Oceania (Table 7). Such large shares indicate, that
the stability of the current banana supply chains is under threat, without considerable efforts in climate change mitigation and
adaptation.

**Table 7: The values presented in the table are in percentage and represent the impacted banana production in different regions**
**across major and complete destructed levels under the 100-RP for both current (1980-2018) and future climates (2015-2050). This**
**table focuses on selected banana-producing regions and acknowledges that not all regions are affected by tropical cyclones. The**
**presented numbers in the table represent shares or proportions rather than absolute values. To provide a more comprehensive**
**understanding, absolute numbers in tons can be found in S3 as well as the % damaged banana producing area per countries.**





| Region | Major | | Complete | |
|---|---|---|---|---|
| | Present | Future | Present | Future |
| Caribbean | 19.5 | 29.2 | 68.1 | 58.4 |
| Central America | 12.4 | 8.9 | 0.0 | 1.3 |
| South America | 1.0 | 1.0 | 0.0 | 0.0 |
| West Africa | 0.2 | 0.2 | 0.0 | 0.0 |
| East Africa | 3.2 | 2.9 | 0.0 | 0.8 |
| South Africa | 0.0 | 0.2 | 0.0 | 0.0 |
| Middle East and North Africa | 10.0 | 15.5 | 0.0 | 0.0 |
| East Asia | 75.5 | 66.1 | 3.0 | 13.3 |
| South Asia | 42.0 | 42.3 | 0.7 | 0.0 |
| Southeast Asia | 38.6 | 43.4 | 7.9 | 8.8 |
| Oceania | 19.1 | 20.5 | 0.0 | 0.9 |


Table 6 provides an overview of the observed variations in impacted banana production between the present and future scenarios. Notably, under (RP50) and the RP100, a substantial increase is observed in class 3, while significant increments are evident in class 1 and class 2 under the RP10. Conversely, a decline in production is seen under the RP200. These changes can be attributed to the intensification of TCs in a warming climate, leading to alterations in their trajectories and a higher likelihood of targeting different areas for impact.

**Table 6: Increase of globally damaged banana production between present (1980-2017) and future (2015-2050) climate conditions (in %) across different damage levels and return periods**


| | RP10 | RP100 | RP200 |
|---|---|---|---|
| Moderate | 20.2 | 12.7 | -6.4 |
| Major | 30.2 | 0.4 | 6.1 |
| Complete | 0.0 | 30.1 | -3.5 |

Interestingly, we observe a slight decrease in damage under RP100 and R200 for the moderate and complete damage classes in some regions (Table 6). This is predominantly due to a decrease in damage in south Asia (India), which, in turn, is driven by a decrease in TC wind speeds at the RP200. This decrease is caused by a projected landward shift in the TC genesis region in the Bay of Bengal. Consequently, TCs are projected to form closer to the coast, meaning there is less time for a TC to intensify to higher wind speeds before it makes landfall and weakens again (see also Figure 2 in Bloemendaal et al 2020a). This results in a decrease in wind speed at higher RPs, leading to lower banana plantation damages in this region.



## 4. Discussion

### 4.1 Impacts on banana production

Despite the clear evidence on the severity of TC impacts on banana production, so far, global scale analyses have not been conducted (Varma, 2019, Malek, 2021). Therefore, our study presents a necessary contribution to increasing the adaptation in the banana supply chains. There is one red thread prevailing throughout our results; the banana producing areas are at risk to be affected by future TC's, which is especially valid for some of the major banana producing regions (Caribbean, South, East and Southeast Asia). According to the IPCC report (2021), it is expected that global temperatures will rise, and heavy fluctuations and extreme weather events are more likely to occur. The effect of these constantly changing weather conditions are an indirect topic of research in this paper. Knutson et al (2020) concluded that there is an increase in intensity of a TC associated with climate change. The results of this analysis are in line with this observation and show a larger share of globally damaged areas at risk and higher damage levels in the scenario with a higher SST. In addition, we did not consider indirect impacts of TCs on banana production. Waterlogging could lead to root rot, and excessive water could lead to favorable conditions for pests and diseases (Aguilar, 2003). In addition, TCs can impact electricity, refrigeration and road infrastructure, leading to disruptions in the supply chains and potential inability to harvest and store the crops (Koks et al. 2019).

### 4.2 Capturing tropical cyclone characteristics

We performed a global-scale spatial analysis, where we applied a generalized procedure to estimate direct physical damage and account for heterogeneity by merging the data into homogenous classes. Analysing hazard damage on a supra-national scale makes validation of the model difficult (de Moel et al, 2015). The IPCC (Koks et al. 2019) concluded that global modelling errors are associated with limitations in parametrizations. We used the STORM model of Bloemendaal et al. (2022), which was demonstrated to perform well on frequency, intensity, and track characteristics, but with some limitations. First, the STORM wind module does not take elevation effects into account. This means that TCs windspeeds can be over- or underestimated over land, as wind speeds can increase or decrease depending on the orientation of for instance a mountain ridge (Bloemendaal et al 2020a). Second, STORM uses a lower bound at 18m/s wind speed (Bloemendaal et al. 2022). However, it was found that banana plants can already be damaged by speeds above 10 m/s, meaning we cannot assess damages up to 18 m/s wind





speed. Our results therefore potentially underestimate damages to banana production. Third, STORM models TCs on a basin scale, meaning that there is no transition of TCs across basins. This can be problematic in regions that

can be affected by TCs originating from multiple basins, such as Central America. Lastly, TCs can also cause damage to banana plantations through excessive rainfall, runoff and landslides. These factors are, however, not modelled in STORM. Inclusion of such factors can potentially alter the risk estimates that were presented in this study.

**4.3 Vulnerability of banana production systems**

We used an empirical approach to constructing the vulnerability curve using a meta-analysis of 22 studies, to increase accuracy of the curve. However, limited data availability can impact the results, and in turn the results can be difficult to transfer in space and time. According to Rhee (2016) much research applies subjective and variable judgment when performing damage assessments (f.i. the value, quality of construction, and type of terrain). Consequently, many damage levels can be considered inconsistent and subject to bias. The vulnerability curve that

we developed is based on 22 TC events damaging banana plantations and expert knowledge. The curve is based on the understanding that all banana producing areas have the same features irrespective of the location or country's political or economic situation. We did not include assumptions on the type of banana, wind, or plantation. However, the impact of wind speed to the banana plant can depend on a variety of factors. Firstly, the type of wind can either positively or negatively influence the productivity of the plant (Robinson, 2010).  Secondly, the

vulnerability of the fruit can depend on the type of banana plant (Coltro, 2019). There were no records on the phenological character of bananas damaged by TCs. An example of this is the Cavendish type, which accounts for nearly half of the total worldwide banana production (CIDAD, 2021). This genotype is less vulnerable to TCs because of their short stems and recovery period. The Cavendish type can produce high yields per hectare and currently accounts for almost half of the total production globally (FAO, 2017). Thirdly, the type of the plantation

identified in the grid cell can vary (Coltro, 2019). Bananas that are produced in a mixed crop system have a larger resilience to TC damage, than those produced on a monoculture farm and should therefore be evaluated separately (Huigen 2006, Robinson, 2010, Mohan, 2013).  Next, it should be noted that because the constructed vulnerability curve was based on a meta-analysis of reported damages in academic and grey literature, there can be a bias towards the more damaging events as these could have been covered more in literature.

Furthermore, factors like population size, income and technology are related to the susceptibility of a country and could also be included to provide a more comprehensive understanding of future TC impacts to the global banana production (Islam et al., 2016). The regional analysis proves to be highly informative as it reveals significant





anticipated increases in damages within key banana-producing regions, namely the Caribbean, East Asia, South Asia, and Southeast Asia. Such areas are expected to face considerably larger impacts on banana production. For example, the share of income generated by banana production is considerably higher in countries like Dominica compared to more industrialized economies like Australia (Beer, 2013, Mohan, 2017). Extreme weather is expected to be even more influential in vulnerable areas where food security is not perceived to be a presumed convenience (Wheeler, 2013, UNICEF, 2022). The impact of natural hazards is of strong influence in low-income countries, where direct losses are generally higher than in high-income ones (Botzen et al. 2019). However, also at country scale, vulnerability assessments may overlook significant urban-rural differences, necessitating the future development of detailed vulnerability curves to accurately simulate storm damages and account for variations in exposure and vulnerability across regions (Baldwin, 2023). We only included physical damage parameters in the development of the vulnerability curve, meaning that future research should also focus on socio-economic and cultural aspects of banana producers. To achieve this, future research should focus on localscales and thereby increasing the accuracy of the vulnerability curves by including more input parameters, among other local terrain and soil characteristics, production system, and implemented adaptation. Such case-studies can consider the climatic conditions and reliance on banana production at the specific location, as well as socio-economic situation of the countries involved. Lastly, future studies could also research low probability events as they often show higher damage levels. Even though we already indicate high damage levels under high probability RPs, the effects of the low probabilities will be even larger and should be identified (Ward et al. 2011).

## 5. Conclusion

Banana producing regions globally are at increasing risk to be impacted by future Tropical Cyclones (TCs). We identify most impacted banana production per region at risk to TC's for the current and future climate under different return periods. This way, the study can help to point out vulnerable areas and consequently be used as a starting point to target and implement adaptation and mitigation measures to reduce the impact of future TC. For the regions where bananas are currently cultivated, adaptation and mitigation measures need to be implemented to reduce TC induced damage, which could also mean changing to a different crop, that is more resilient to TCs. To establish more resilient banana supply chains, this analysis can serve as a tool to identify risky regions and invest in regions with a low probability of TC induced damage. Overall, we highlight the need for further research and implies action is to be taken to mitigate the risks of future TCs to banana producing areas, let alone to protect the livelihoods of those dependent on this important commodity. Studying the effects of extreme weather events to





agricultural commodities is important as it provides us with a long-term outlook that considers our current and future generation. However, the rising and overarching challenge to deal with extreme weather and its impact to agricultural commodities is one that requires assessing climate change impacts across the whole supply chain.


**Competing interests**

The contact author has declared that none of the authors has any competing interests.

**Acknowledgements**

MCdR received support from the MYRIAD-EU project, which received funding from the European Union's Horizon 2020 research and innovation programme under grant agreement No 101003276, and by the Netherlands Organisation for Scientific Research (NWO) (VENI; grant no. VI.Veni.222.169).




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
