# Peer review of "Assessing future impacts of tropical cyclones on global banana production"

_Natural Hazards and Earth System Sciences, 2023_

## Author Comment (AC1)

**Reviewer 1**

**Dear authors,**

Thank you very much for your work on this manuscript. I think the direction of the study is very important: TCs and bananas are a key research topic, and your work here fills an important research gap.

We would like to thank the reviewer for their very kind and constructive comments. Please see our replies to the reviewer's comments below.

I do have several comments to help to improve your manuscript, as there are several aspects of the manuscript that need work.

For example, you mention expert knowledge, but refer instead to a meta-analysis, and this is also referred to as empirical work?

We fully agree with the reviewer that this was unclear. In the methods section we have made significant changes in the overall structure and tried to improve our explanation of the approaches used for the different elements of risk. In section 4.3 we have also adjusted the order of the text of the paragraph to improve its flow. The rewritten part of section 2.3.1 now reads as follows:

To construct the curves, we use an approach similar to the research by Yum et al. (2021) (see Figure 2) where the authors use existing white and grey literature reporting on historic TC events, their maximum wind speed and damages. While we acknowledge that TC impacts are often caused by an interplay of wind, precipitation and storm surge, we use wind speed as a proxy for all three hazards as information on the latter two hazards is often lacking or incomplete (see for instance Eberenz et al., (2019) and Yum et al., (2021) for a similar approach for building vulnerability curves). Such an approach based on empirical data allows for a higher accuracy of the damage curve (Merz et al., 2010).

There are also several inconsistencies and the discussion is far too broad to be impactful. Who are you writing for? Who would you like these results to aid? While the results are relevant I struggle to see an effort to make a connection to readers or stakeholders. For example, there is only one sentence that describes the actual impact of a hurricane on banana production in Honduras. Taking a risk-based approach, there is certainly greater exposure and vulnerability to TCs in the Western North Pacific Basin, where the Philippines is a major grower and greatly affected by TCs annually, with some of them making severe impacts to banana production (e.g. Typhoon Bopha).

We fully agree that this was less clear in the earlier version of our manuscript. Our research aims to provide general insights into the (future) impacts of TCs on as important a crop as bananas. As this is a very interdisciplinary topic, our findings can be useful for other researchers from both the TC and the crop research fields. But it can also benefit international organisations such as the FAO in (re-)directing adaptation funds. We have tried to clarify this in different parts of the manuscript. For example, in the conclusions we now write:

Our study results can support the identification of vulnerable areas and can therefore be used as a starting point to target and implement adaptation and mitigation measures, such as awareness raising efforts and early warning systems. This will hopefully contribute to a reduction in the impact of future TCs to banana plantations.

I also make a point to be more geographically specific in your analysis. Saying that "Asia" is more impacted is an extremely broad statement, as it is a diverse continent. Which countries in Asia are banana growers, and which ones will be impacted?

We fully agree and have not only increased specifics of geographical locations but also of results.

While I appreciate the scientific and technical work behind this paper, please see my comments as hopefully helpful revisions to improve the presentation of your results. What are the steps forward for banana growing? How can adaptation efforts utilize these modelling studies?

With the very helpful comments from the reviewer, we have tried to improve the presentation of our results. In the conclusions section, we have provided examples of how our results can support adaptation efforts.

Thank you for your work, and I'm available to revise your work again.

**Minor text notes: Reference style is somewhat inconsistent (with commas, no commas, semicolon, etc.). Please make this consistent throughout the document.**

We thank the reviewer for pointing this out. We have critically revised and homogenized the reference style throughout the manuscript.

Also, I would suggest line numbers rather than paragraph numbers to aid the reviewer.

We understand the confusion, however the journal automatically adds line numbers by increments of 5.

**Abstract**

**Please minimise the use of acronyms in the abstract to aid the unfamiliar reader**

We have critically revised the use of acronyms in the abstract and have reduced the number of abbreviations.

**What does 'majorly' mean in this context? Is there a way to quantify this in the abstract?**

We thank the reviewer for pointing this out, and we followed up on their suggestion and rephrased and quantified the term to "almost (> 79%)".

'Asia' is a whole continent, not a region (e.g. compared to the Caribbean region). Be more specific here.
In line with earlier comment sand replies, we have tried to address this as much as possible.
Para 25: I would suggest that the text opens with the importance of bananas as a crop first, before going to climate change affecting it.

We have made substantial changes to the introduction which now reads as below. For now we left the order unchanged, but we will critically reflect on switching the order of the first two paragraphs to change the perspective from TCs to bananas.

[revised manuscript text omitted]

Para 25: Coumou 2012 reference – is this not an outdated reference, considering that the IPCC released a synthesis report in 2023? There has also been significantly more literature on tropical cyclone projections since. This may need an update as it appears several times in the introduction.

We fully agree and have adjusted this.

Para 25: "Particularly..." is not a complete sentence.

We checked the sentence, and we believe it is alright.

**Para 25: categories?**

This was an oversight and we have removed this.

**Para 25: "CC"?**

This refers to Clausius-Clapeyron, but we have rewritten the entire sentence.

Paras 35 and 40: I would also emphasise the socio-economic importance of the crop, and for farmers' or communities' livelihoods more than just consumer prices. This appears briefly in Para 55, 110, 115.I think the introduction section could be tightened up in terms of its flow and writing as the information presented is a bit scattered throughout the first few sections.

We agree on this point and have critically rewritten the introduction and checked this throughout the rest of the manuscript.

Para 45: Recent analyses from the Philippines shows more data to support these paragraphs on major TC damages to banana production, and that recovery can be up to 5 years post-TC: e.g. <a href="https://doi.org/10.18783/cddj.v005.i01.a05">https://doi.org/10.18783/cddj.v005.i01.a05</a>; <a href="https://journalofnaturestudies.org/files/JNS19-2/62-83\_Damasa-Macandog\_Assessment\_Impacts\_Flashflood.pdf">https://journalofnaturestudies.org/files/JNS19-2/62-83\_Damasa-Macandog\_Assessment\_Impacts\_Flashflood.pdf</a>; <a href="https://www.ijisrt.com/assets/upload/files/JJISRT21DEC732.pdf">https://www.ijisrt.com/assets/upload/files/JJISRT21DEC732.pdf</a>

Para 120/125: What are the limitations of this assumption re: equal vulnerability of all banana cultivars? Do your datasets differentiate between different types of banana cultivars? As you are using plantation data, this are likely to be monocultures of the Cavendish variety. Please elaborate on this assumption as it is important to distinguish what farm type is considered in the analysis. We would like to clarify that we are looking at very destructive events so damage will be similar and there is no systematic data. Based on the (limited) vulnerability literature on this topic, we make the assumption that for TCs there won't be a significant difference between banana crop types.

**Para 130: A reference missing for the final sentence.**

Thank you, we have address this.

Para 155: it is unclear if you, in this work, employed expert opinion. Relying on a review of literature is not the same as soliciting expert opinion. (cf. para 180). This has impacts on your description of methods to determine damage categories (cf Table 2). You also later describe this is as a 'meta-analysis' (cf. Figure 1).

We agree this was unclear, we have explained how we addressed this in an earlier reply.

Para 230. 's.'? Thank you, we have address this.

**Para 230-235 'areas' – banana growing areas? Areas in general? What is a 'banana area'? Be more specific.**

We fully agree and now write "banana producing area".

**Figure 2, Table 5: future climates implies several scenarios, but I believe you only used SSP585. Please clarify throughout the text.**

Thank you, we have addressed this.

Figure 2: why is the map cropped? Please show the whole map. There are also banana-growing regions in the subtropics, and I don't know (or you haven't described) if the IFPRI dataset is limited to tropical areas only.

We understand this comment. The reason it looks like that is because we wanted to use a non-colonial map projection. To improve the clarity of our results, we only show areas where tropical cyclones occur. We found it challenging to visualize 6 different global figures on the impacts of TCs on banana cultivation. Showing the whole map does not provide additional information, as areas outside the cropped map are minor. We provide full maps in a higher resolution in the supplementary material, where each RP for the baseline and future scenario are displayed in a larger format.

**Table 4: what is a 'globally damaged banana producing area'?**

Thank you, we meant area and we have address this.

Para 255: as with the abstract, please be more specific with your language. Southeast Asia versus 'Asia' broadly allows for a more accurate representation of banana growing in the region, which is highly concentrated in the Philippines at the moment – this is also a country highly vulnerable to TCs. I think it would be helpful to ground your analysis (or maybe just introduction) in more practical examples of TCs and their impacts.

Countries in Asia that will be highly impacted are China India and several countries in southeast Asia. The most impacted countries in southeast Asia are: Vietnam, Philippines, Cambodia and Indonesia. In the text we now write Southeast Asia, China and India instead of Asia.

**Para 270: what does this first sentence mean, exactly?**

Area is not production totals/share in global production; we have clarified this in the text.

**Para 280, 315: 'We found out', 'especially valid' are quite journalistic phrasing – please make this more a report of results.**

We agree and we have adjusted this.

Also, same comment about broad reporting of the 'Asia' region. This paper would be more valuable if you named the specific countries rather than the whole continent. We agree and have addressed this.

Para 315: banana supply chains implies the whole system of food supply, including transport, storage, and distribution. Do you mean this? Or just production? We have rewritten and clarified this.

Para 325: pests and diseases such as...? Banana pests and diseases are a major research point in banana production, and it is only briefly mentioned in passing here.

We explain in more detail how our study underestimates potential impacts due to not including indirect impacts such as increases in pests and diseases.

Para 335: "The IPCC (Koks et al. 2019) concluded that global modelling errors are associated with limitations in parametrizations." – what are you trying to say by adding this citation here? I think your discussion of modeling limitations in this paragraph is very valuable, but this statement feels like a blunt addition, and could be made more specific in the context of your study limitations. We agreed and have addressed this.

Para 350: I would revise and review this paragraph. Was it an empirical approach? Was it expert knowledge? Which experts were consulted? Or was it a meta-analysis? The lack of clarity in methods on this point weakens the paper.

We agree with the reviewer that this was unclear. In the methods section (2.3.1) we have improved our explanation of the empirical approach used. In section 4.3 we have also adjusted the order of the text of the paragraph to improve its flow.

Para 360: "This genotype is less vulnerable to TCs because of their short stems and recovery period." – I'm not sure about this. What is your reference here? Cavendish bananas are typically planted in monocultures and the landscape homogeneity can contribute to its vulnerability to hazards.

We rewrote the section on genotypes and removed some information. We now focus more on the type of plantation.

**Para 360: "An example of this" – an example of what, sorry?**

We solved this when addressing the previous point.

**Para 370: what do you mean by "the susceptibility of a country"? Some of these statements are unfortunately exceptionally broad.**

We would like to clarify that we focus on the (relative) "short-term" vulnerability analysis (ie impacts on banana crops right after extreme TC event) rather than the long-term vulnerability of a country or banana production chain (e.g. how resilient is the country, can it easily aid speedy recovery of the banana plants). We tried to clarify this in the text.

Paras 370-390: These statements/ citations are simply placed together, with little additional insight. Why is there a comparison to Australia? Why are parameter limitations re-discussed here? How can the results of the study help to inform adaptation in agricultural production? Can they be connected to other efforts to understand food security under changing TCs?

We agree that this wasn't clear, and we have adjusted this whole part of the results section.

**Reviewer 2**

This is an interesting paper looking at the impacts of tropical cyclones on banana production under climate change. The most novel aspect is the vulnerability curve; I think how it was obtained needs to be explained better. Another major technical concern is that it appears that the maximum sustained wind speed is assumed to be experienced at the plantation when surely that is not generally the case (and if it was not done this way, then the methodology is not explained adequately) --- this affects the vulnerability calculation as well as the impacts projections. I have some other comments listed below. I think the paper needs major revision.

We thank the reviewer for their kind and very valuable comments on our manuscript. Below we address each of their comments separately and show the changes we made in the manuscript to address their points.

**Major comments:**

The main innovation here seems to be the development of the vulnerability curve. Given this, it is not described in enough detail. The reader should be able to reproduce it if they wish to. Data and code should be provided. What is the functional form assumed for the curve? How are the values interpolated between the small number of categories for which damage was calculated explicitly? Etc.

We thank the reviewer for their comment and fully agree that this was not explained well. We have made significant changes to this section, trying to better explain this. The curve follows the typical wind-vulnerability curve shape and we have included a more thorough explanation of how it was derived and included the excel file that shows how the curve was made. We did note that in developing the original appendix, some strange shift in numbers took place. However, the curve appears to be correct. Unfortunately, no similar curves exist for bananas so we cannot validate it with other curves. We will very critically reflect on the original appendix and adjust things accordingly.

Along with the above, some representation of uncertainty should be given. The results are stated to a high degree of precision, 2 significant figures in the percentages. Surely this level of precision is not justified given the many uncertainties.

We agree that there are many uncertainties. We checked the writing of numbers carefully and have used up to a max of one decimal behind the comma throughout our manuscript.

In supplemental table 1 only the maximum sustained wind speed for the storm is listed. Was it assumed that this wind speed was experienced at the banana plantation? Surely this is not correct unless the storm hits very directly, and maybe not even then. A parametric wind profile should be used to estimate the wind speed at the plantation. Please explain and justify better what was done. This goes both for the development of the vulnerability curve and the subsequent calculation of impacts.

As explained in an earlier reply, we noticed that in creating the original appendix 1, some strange shift of numbers occurred. We will look into this very critically and then in depth address this point of the reviewer.

One comment that is not about the results themselves, but how the results are communicated: there are many statements (including in the abstract) of the form that X percent of banana producing areas will experience some level of damage at 100-year return period. I think this is misleading. I believe what is mean that X percent will be damaged if they experience a 100-year

event at their location. But the way it is stated sounds like a 100-year event for the planet will see X percent of banana production damaged at once --- which is not true, because the different areas will not experience 100-year storms at the same time. Please clarify this everywhere it comes up, including the abstract.

We agree with the reviewer and have rewritten and clarified this in the text.

**P14: the area around the Bay of Bengal seems important to these results. My understanding of the STORM dataset though is that it has a poor representation of storms in this region due to its neglect of wind shear. Please clarify if this is the case and caveat appropriately.**

The reviewer is right in their understanding of the limitations of the STORM dataset. We have added a discussion on this (and some other limitations) in the Discussion (Section 4):

There are also limitations in the STORM model that could affect the outcomes of this study. First of all, TC intensity in STORM is solely modelled as having a direct relationship with sea-surface temperatures. In reality TC intensity is also largely influenced by vertical wind shear; an effect that is absent in STORM. While in general STORM validates well against observations (Bloemendaal et al., 2020c), there are regions where vertical wind shear in reality plays a critical role in governing TC intensity. One of these regions is the Bay of Bengal; for this region, TC intensity and associated RPs tend to be over- and underestimated, respectively. As a consequence, our impact assessment for this region can be overestimated. Secondly, STORM uses the parametric wind field model from Holland (1980) to translate point data to a 2-dimensional wind field. This model assumes asymmetry in the wind field to arise from background flow; in reality, these asymmetries can also be induced by enhanced wind shear or interaction with land. This may result in slightly altered wind speed RPs. Lastly, TC decay after landfall is modeled through an empirical inland decay function (Kaplan and DeMaria, 1995). This decay function was derived based on USA landfalls, and hence may perform less well elsewhere.

**The discussion of caveats at the end is very long and detailed, more so than the explanation of the calculation of vulnerability itself (which is surely a source of great uncertainty). This seems a bit unbalanced.**

We fully agree and have made significant changes in this section to address this. The discussion section now reads as follows:

[revised manuscript text omitted]

**The last paragraph of the paper is a bunch of boilerplate statements that seem unnecessary. I suggest sticking to summarizing the findings of the research itself.**

We fully agree and have made significant changes in this section to address this.

**Minor/editorial comments:**

**P1, I2 of main text: Coumou, 2012 reference for the global mean warming is a bit odd. It's 12 years old, and an 0.5 degree uncertainty seems way too large. Please use an updated reference and a more precise number, this is a very well-studied quantity.**

We agree with the reviewer and have critically revised the introduction. The sentence now reads:

Currently, the average global surface temperatures haves already increased between 1.1 and 1.2 degrees Celsius above pre-industrial levels (IPCC, 2023).

**P1, last line: what does "these categories" refer to?**

This sentence has been removed from the introduction.

**Some references (e.g., Baldwin, 2023) are not in the reference list. Others (e.g., Robinson) seem to have incomplete information. Please do a detailed check on all of them.**

We thank the reviewer for pointing this out. We have critically revised the use of references and have removed or updated references from the reference list.

**P2, I30: but this decrease in Walker circulation and trend towards El Niño has not been observed, and there is currently a very active debate about it, so it seems inappropriate to cite it uncritically.**

**P2, I34: what is "CC"? This needs more explanation.**

We agree with the reviewer and are aware that there are active debates on the ENSO bias in climate models. We have decided to remove this entire paragraph from the Introduction. As this paragraph has now been omitted, the term "CC" has also been removed.

**P2, I45: is this really true? How does the FAO determine this? It seems quite surprising indeed given how many buildings TCs destroy, etc.**

The FAO report we cite, states how a recent tropical cyclone decreased production and exports in Guatemala, Mexico, Honduras and Philippines. In Honduras for example, 40% of plants were destroyed in 2021, and in the Philippines exports were lower by 37% due to a hurricane. All this leads to a worsened economic situation for tens of thousands of people in the mentioned countries.

The rewritten introduction (as shown in an earlier reply to reviewer 1) reflect our more careful wording on this.

**P2, I57: what does "these indices" refer to? And is "mitigation or adaptation" really meant seriously? Surely whatever banana growers can do to mitigate climate change (i.e., reduce their own emissions) will have a negligible impact on their own impacts from climate change, and any real direct benefit to them can only come through adaptation. This is repeated again in the conclusions, line 396.**

We have rephrased that part of the sentence so that it now reads "the likelihood of TC impacts". We also agree with the reviewer that we should focus on adaptation rather than mitigation strategies. We have carefully revised the manuscript and removed the use of the word mitigation wherever appropriate.

P4, I103: there may be little difference between scenarios by 2050, but it becomes much larger later. This should be made clear. We thank the reviewer for pointing this out, and have added a paragraph on this in the Discussion (Section 4):

The STORM future climate datasets were generated based on the high-end SSP585 scenario. While one can discuss the likelihood of this emission scenario, as current developments are steering away from this scenario, the average climate conditions over the 2015 - 2050 time period do not differ substantially between the different forcing scenarios. We therefore believe that the SSP585 input dataset as was used for Bloemendaal et al., (2022) can be seen as a good proxy for changes in TC characteristics over the aforementioned time period. While this study does not consider TC impacts beyond 2050, we alert readers that the average climate conditions past 2050 do start to deviate and that the approach as used in Bloemendaal et al., (2022) does not hold then.

**P6, l153: this citation to Yum (2021) is not sufficient, please explain exactly what was done (see major comments above).**

We agree and as replied in earlier comments to Reviewer 1 and 2, we have tried to address this.

**P7, 1183: as in major comments, please clarify how wind speed at each plantation was obtained. It appears here that it was assumed to be the maximum sustained wind for the storm, but surely this is incorrect.**

We have replied and reflected on this in an earlier reply.

**Table 2: please give units for wind speed (presumably m/s).**

Thank you, we have addressed this in the table.

---

## Author Response (AR2)

**Rebuttal round 2 of TCs and bananas**

**Reviewer 1**

Dear authors, I think that the revision adequately addresses my concerns and comments and presents a more specific and coherent analysis of risk and vulnerability. There are however some leftover minor errors, such as a "(??)", empty or erroneous references that should be corrected and proofread thoroughly prior to submission.

We thank the reviewer and have addressed these oversights.

**Reviewer 2**

The paper is improved, but still seems to me to draw misleading inferences from the results. (The reviewer checklist, by the way, does not really ask about this --- whether the top-line conclusions are framed in a way that is justified by the actual results --- which is why my ratings of "Good" are a bit dissonant with my complaints here.) The uncertainties are under-recognized and the conclusions drawn do not make sense to me given those. It seems to me that the changes found are much smaller than the real uncertainties, but this is not acknowledged. The revisions I suggest are minor, in the sense that they wouldn't require much work to implement, hence my rating of minor revisions. (Also, I don't really want to review it again; the editor can easily determine if the authors have followed my suggestions or not.) But I think it would be a mistake to publish the paper as is, without modifying the headline conclusions statements to reflect the uncertainties; and within the paper, documenting those uncertainties a bit more.

My most important, large-scale comment: The abstract concludes "Globally, 24.3% of all banana-producing areas are projected to suffer major or complete (>84%) damage under current climate conditions, increasing to 26.5% under future climate scenarios at the 100-year wind speed return period. Additionally, we estimate that 30.1% of global production under current conditions, and 31.1% under future conditions will be majorly or completely damaged at the 100-year return period. The regions predominantly affected in the future are Southeast Asia, China, India and the Caribbean, potentially experiencing substantial disruption in banana production. Our results therefore indicate that considerable efforts in climate adaptation are essential ensure the stability of chains." to global banana supply

However these changes (24.3->26.5% and 30.1->31.1%) are very small. Surely they are well within the uncertainties, so that we can't be confident about even the sign of the change! In fact it seems slightly absurd to me to give these results to three significant

figures. And some regions show decreases in risk; however this could all be different if a different TC model were used (some have increases in TC frequency, some decreases, etc.). Altogether the framing and conclusions seem an exercise in false precision.

We agree and have addressed this by also communicating the range of changes, and by adding an explanation that the increase is "on average". We also list the largest regional mean differences.

A few more specific comments:

1. The explanation of the derivation of the damage function is better now, but I would still like to understand the uncertainty. How about plotting the 22 data points on Fig. 1, as a scatter plot on top of the curve? This would add no cost and give the reader much more insight into the the most original aspect of this paper.

We fully agree and have addressed this.

2. The authors say (lines 177-8) "evidence of damage to banana plantations was already observed at windspeeds of 10 m/s, hence why we set the lower limit of the curve (Vmin) at 10 m/s." In fact fig. 1 shows that damage function begins at zero, so that any wind at all destroys some bananas, and at 10 m/s already around 50% are destroyed. This is hard to believe, since these wind speeds occur all the time. With this curve, in fact, it wouldn't make sense only to use TC tracks as the source of winds, as routine daily weather phenomena would also cause damage and this should be included. Please address this in some way.

We agree with the concern raised by the reviewer and we have adjusted the curve accordingly.

3. Please say something about how STORM uses the climate models to get climate change influenced storms. I know it's in other papers but a sentence or two of explanation would really help the reader. Some discussion of the uncertainties would be very welcome as well. Can we expect that we might get different results if another TC model were used? I think so! This is relevant to the first, major comment above.

We agree and have added the following:

"The STORM future climate synthetic datasets are constructed using information from baseline (1979 – 2014) and near-future (2015 – 2050) climate model runs (SSP-585) for each of the different general circulation models. TC statistics (frequency, intensity) are

extracted from both the baseline and near-future runs, and changes in these statistics (the so-called "delta") are calculated. Subsequently, these deltas are added to the historical dataset which was used as input to create the STORM historical dataset. This creates a "future climate version" of this historical dataset, without the biases often present in general circulation models. For more information on this method, we direct readers to Bloemendaal et al., (2022)."

---

## Author Response (AR3)

Dear Ira,

Many thanks for your feedback. We have now included elaborate discussions of uncertainty regarding both the hazard and vulnerability aspects. You will find this in the relevant chapter, the discussion, and the conclusions. We have also addressed the issue of "some damage" and rephrased this to moderate to severe damage.

Many thanks,

Best,
Marleen de Ruiter